# Time Trends in Ischemic Heart Disease Mortality Attributable to PM_2.5_ Exposure in Southeastern China from 1990 to 2019: An Age-Period-Cohort Analysis

**DOI:** 10.3390/ijerph20020973

**Published:** 2023-01-05

**Authors:** Weiwei Wang, Nan Zhou, Hao Yu, Huafeng Yang, Jinyi Zhou, Xin Hong

**Affiliations:** 1Department of Non-Communicable Disease Prevention, Nanjing Municipal Center for Disease Control and Prevention, 3 Zizhulin Road, Gulou District, Nanjing 210003, China; 2Department of Epidemiology and Biostatistics, School of Public Health, Nanjing Medical University, 101 Longmian Avenue, Jiangning District, Nanjing 211166, China; 3Department of Non-Communicable Disease Prevention, Jiangsu Provincial Center for Disease Control and Prevention, 172 Jiangsu Road, Gulou District, Nanjing 210009, China

**Keywords:** PM_2.5_, ambient PM_2.5_, household PM_2.5_, ischemic heart disease, age-period-cohort analysis

## Abstract

PM_2.5_ exposure is a major environmental risk factor for the mortality of ischemic heart disease (IHD). This study aimed to analyze trends in IHD mortality attributable to PM_2.5_ exposure in Jiangsu Province, China, from 1990 to 2019, and their correlation with age, period, and birth cohort. Methods: Data were extracted from the Global Burden of Disease study 2019 (GBD2019). The magnitude and direction of the trends in IHD mortality attributable to PM_2.5_ exposure were analyzed by Joinpoint regression. The age-period-cohort (APC) model was used to evaluate the cohort and period effect. Results: Age-standardized mortality rate (ASMR) of IHD attributable to PM_2.5_ exposure decreased from 1990 to 2019, with an average annual percentage change (AAPC) of −1.71% (95%CI: −2.02~−1.40), which, due to ambient PM_2.5_ (APM) exposure and household PM_2.5_ (HPM) exposure increased with AAPCs of 1.45% (95%CI: 1.18~1.72) and −8.27% (95%CI: −8.84~−7.69), respectively. APC analysis revealed an exponential distribution in age effects on IHD mortality attributable to APM exposure, which rapidly increased in the elderly. The risk for IHD mortality due to HPM exposure showed a decline in the period and cohort effects, which, due to APM, increased in the period and cohort effects. However, favorable period effects were found in the recent decade. The overall net drift values for APM were above zero, and were below zero for HPM. The values for local drift with age both for APM and HPM exposures were initially reduced and then enhanced. Conclusion: The main environmental risk factor for IHD mortality changed from HPM to APM exposure in Jiangsu Province, China. Corresponding health strategies and prevention management should be adopted to reduce ambient air pollution and decrease the effects of APM exposure on IHD mortality.

## 1. Introduction

Particulate matter pollution with a diameter of less than 2.5 μm (PM_2.5_) is the fourth leading risk factor for global mortality [1]. Two main forms of PM_2.5_ have been quantified in the Global Burden of Disease (GBD) 2019, namely, ambient particulate matter (APM, exposure to PM_2.5_ in the outdoor air) and household particulate matter (HPM, exposure to PM_2.5_ due to solid fuel use). They present correlations with socio-demographic development, in which APM exposure increases in the low-to-moderate socio-demographic index (SDI), while HPM steadily decreases with socioeconomic development [2]. The age-standardized mortality rate (ASMR) attributable to PM_2.5_ exposure is extremely high in Asian countries. It is reported that PM_2.5_-exposure-induced health issues changed from global public health concerns to those that mainly affect low- and middle-income countries [1,3,4,5]. In China, PM_2.5_ exposure ranks the third leading risk factor for all-cause mortality, causing 1.79 million deaths per year [1].

Ischemic heart disease (IHD) is a major public health problem worldwide. It is the leading cause of global mortality, resulting in 9.14 million deaths and 182 million disability-adjusted life years (DALY) in 2019 [1]. ASMR for IHD has substantially declined in the past three decades throughout the world, while it is estimated as predominantly increasing in South, East, and Southeast Asia, including China [2]. In 2019, about 20% deaths (1.87 million) in China are caused by IHD, which was the second leading cause of death [6]. PM_2.5_ exposure is a well-defined risk factor for IHD. Long-term or short-term PM_2.5_ exposure is strongly associated with an increased mortality risk for IHD [7,8,9,10]. A 10-µg/m^3^ increase in long-term average PM_2.5_ exposure results in a 23% increase in IHD mortality and 24% increase in cerebrovascular disease mortality [11]. Moreover, a 10 µg/m^3^ increase in HPM exposure causes a 4–6% increase in the risk of all-cause mortality and a 10% increase in the risk of cardiovascular diseases (CVD) [12,13].

Great efforts have been made on exploring the changes of IHD mortality over time and age-specific mortality. However, the period and cohort effects on IHD mortality caused by PM_2.5_ exposure have not been elucidated [14]. An APC is usually used to determine the varying effects of age, period, and birth cohort, as well as their corresponding coefficients, thus identifying high-risk populations and assessing the efficacy of early interventions on them [15]. In the present study, we analyzed relevant data from GBD 2019 by introducing the APC model, aiming to explore time trends in IHD mortality attributable to PM_2.5_ exposure (APM and HPM) in Jiangsu Province located in the Yangtze River Delta, Southeast China, from 1990 to 2019. Our findings may provide evidence-based management for air pollution control and health resource allocation.

## 2. Materials and Methods

### 2.1. Data Sources

The attributable burden of IHD data in Jiangsu province from 1990 to 2019 was directly obtained from the GBD 2019 at the provincial level, which is scientifically and comprehensively evaluated by the World Health Organization (WHO) and the Institute for Health Metrics and Evaluation (IHME), including sex- and age-specific annual deaths and ASMR of IHD attributed to APM and HPM exposures. The original data of IHD mortality were mainly from the Disease Surveillance Points System (DSPs), the Maternal and Child Surveillance System, and the Cause of Deaths Reporting System of the Chinese Centers for Disease Control and Prevention (CDC) [16]. The medical classification of IHD was made on the basis of the International Statistical Classification of Diseases, tenth revision (ICD-10), with the disease code ranging I20–I25.9.

In the GBD study, there are two main forms of PM_2.5_ exposure, namely, APM and HPM. APM is defined as the population-weighted annual average mass concentration of PM_2.5_ in a cubic meter of air. This measurement is reported in μg/m^3^. The APM was measured by annual average, which was estimated from the integration of satellite observations of aerosols in the atmosphere combined with chemical transport model simulations, surface measurements, and geographical data at a 0.1° × 0.1° (approximately 11 × 11 km at the equator) resolution, and then aggregated to population-weighted means to produce an exposure estimate [1,17]. HAP is estimated from both the proportion of individuals using solid cooking fuels and the level of PM_2.5_ air pollution exposure for these individuals. Solid fuels include coal, wood, charcoal, dung, and agricultural residues. The APM concentrations estimated the standard multi-country survey series such as demographic and health surveys (DHS), living standards measurement surveys (LSMS), multiple indicator cluster surveys (MICS), and world health surveys (WHS), as well as censuses and country-specific survey series. To fill the gaps of data in surveys and censuses, estimates were downloaded and updated from the WHO Energy Database [1]. ASMR of IHD attributable to PM_2.5_ exposure (APM and HPM) was computed on the basis of the World standard population [2].

### 2.2. Statistical Analyses

The magnitude and direction of trend in IHD mortality attributable to PM_2.5_ exposure (APM and HPM) over time were assessed by calculating the average annual percentage change (AAPC) and corresponding 95 CI% via the Joinpoint regression using Joinpoint software (4.9.1.0). The APC model was designed to extract information of mortality to estimate the potential correlation of IHD mortality due to PM_2.5_ exposure (APM and HPM) with age, period, and birth cohort. The age effects were risks linked to outcomes of aging specific to individuals. The period effects were any outcomes linked with living during a particular period in all age groups, including the control and prevention strategies, policies, and regulations. The cohort effects were any changes in outcomes between groups with the same birth years, presumably arising from differences in the lifestyle and exposure degrees to risk factors. The general logarithmic linear form of the APC model is
(1)$$\rho =\alpha_{a}+\beta_{p}+\gamma_{c}$$
where *ρ* is the expected incidence rate, and *α_a_*, *β_p_*, and *γ_c_* indicate the effects of age, period, and cohort, respectively. It can be transformed into the form of age–period (2) and age–cohort (3):
(2)$$\rho_{ap}=\mu +(\alpha_{L}-\gamma_{L})(a-\bar{a})+(\pi_{L}+\gamma_{L})(p-\bar{p})+{\tilde{\alpha }}_{a}+{\tilde{\pi }}_{p}+{\tilde{\gamma }}_{c}$$(3)$$\rho_{ac}=\mu +(\alpha_{L}+\pi_{L})(a-\bar{a})+(\pi_{L}+\gamma_{L})(c-\bar{c})+{\tilde{\alpha }}_{a}+{\tilde{\pi }}_{p}+{\tilde{\gamma }}_{c}$$
where
$\alpha_{L}-\gamma_{L}$ represents cross-sectional trend; $\alpha_{L}+\pi_{L}$ represents longitudinal age trend; $\pi_{L}+\gamma_{L}$ represents net drift; and ${\tilde{\alpha }}_{a}$, ${\tilde{\pi }}_{p}$, and ${\tilde{\gamma }}_{c}$ represent age, period, and cohort deviation, respectively.

The fitted APC model estimated many useful estimable functions. This study mainly focused on the following estimable functions. Net drift, which is the overall logarithmic linear trend by period and birth cohort, was used as an estimate of the overall AAPC for the outcome measure; conversely, the local drift was the logarithmic linear of each age group by period and birth cohort that reflected the AAPC of outcome measures in each age group. A drift of a minimal ±1% per year indicated a significant change in mortality [15]. The longitudinal age curve was plotted to calculate the fitted age-specific rates adjusted for period deviations in a reference cohort, thus reflecting age effects. The period (or cohort) rate ratio (RR) was the ratio of age-specific rates value in each period (or cohort) relative to the reference period (or cohort), thus reflecting period (or cohort) effects.

This study used the analysis tool for APC model developed by the Biostatistics Branch of National Institutes of Health, USA. In the APC model, it was necessary for the collected data to be converted to successive 5-year age groups and consecutive 5-year periods. On the basis of the format of the GBD database (five-year age groups with annual data), we arranged mortality and population data into 14 age groups as follows: 25–29 years group, 30–34 years group, 35–39 years group, 40–44 years group, 45–49 years group, 50–54 years group, 55–59 years group, 60–64 years group, 65–69 years group, 70–74 years group, 75–79 years group, 80–84 years group, 85–89 years group, and 90–94 years group. Individuals younger than 25 years were not included in the present study due to the rare deaths in this age population. The period was divided into 6 consecutive 5-year periods as follows: 1990–1994 period, 1995–1999 period, 2000–2004 period, 2005–2009 period, 2010–2014 period, and 2015–2019 period. According to the relationship between age, period, and cohort, 19 corresponding consecutive birth cohorts were generated by subtracting the age at death from the period of death as follows: 1898–1902 group, 1903–1907 group, 1908–1912 group, 1913–1917 group, 1918–1922 group, 1923–1927 group, 1928–1932 group, 1933–1937 group, 1938–1942 group, 1943–1947 group, 1948–1952 group, 1953–1957 group, 1958–1962 group, 1963–1967 group, 1968–1972 group, 1973–1977 group, 1978–1982 group, 1983–1987 group, and 1988–1992 group, with the central age group, period, or birth cohort as reference groups in all APC analyses. The significance of the estimable parameters and functions was tested by the Wald chi-squared test. Two-sided *p* < 0.05 was considered statistically significant.

## 3. Results

### 3.1. Trends in IHD Mortality Attributable to PM_2.5_ Exposure

The trends in IHD mortality attributable to PM_2.5_ exposure (APM and HPM) are listed in Table 1 and Appendix A. Although the number of IHD deaths due to PM_2.5_ exposure increased from 8636 to 14,462 over the past three decades, the ASMR of IHD attributable to PM_2.5_ exposure decreased in both men and women, especially in women (AAPC: −2.06%; 95%CI: −2.26~−1.85). We further analyzed the ASMR of IHD attributable to APM and HPM exposure. In detail, the number of IHD deaths due to APM exposure from 1990 to 2019 increased by 4.2 times, from 3183 to 13,265. ASMR increased in the whole cohort (AAPC: 1.45%; 95%CI: 1.18~1.71), with the highest ASMR in 2010 for men (19.00/100,000) and women (10.13/100,000). In contrast, ASMR of IHD attributable to HPM exposure showed a remarkable decrease trend in the whole cohort (AAPC: −8.27%; 95%CI: −8.84~−7.69). For men, AMSR of IHD attributable to APM or HPM exposure from 1990 to 2019 was higher than that in women, and ASMR due to APM exposure since 1995 was higher than that due to HPM exposure. For women, ASMR of IHD attributable to APM exposure was higher than that due to HPM exposure after 2000 (Appendix A).

### 3.2. ASMR for IHD Attributable to PM_2.5_ Exposure

The mortality of IHD attributable to PM_2.5_ exposure (APM and HPM) by age, period, and median birth cohorts from 1990 to 2019 is listed in Figure 1 and Appendix A. IHD mortality increased with age, and similar changes were seen in IHD mortality due to APM and HPM exposure in all periods (Figure 1A–C). The declining trend of IHD mortality attributable to HPM exposure was pronounced from 1990–1994 to 2015–2019 in all age groups (Figure 1C). In contrast, IHD mortality attributable to APM exposure gradually increased from 1990–1994 to 2010–2014, and then decreased in 2015–2019 in all age groups (Figure 1B). The mortality of IHD attributable to HPM exposure declined sharply across birth cohorts (Figure 1F), whereas that attributable to APM exposure was initially increased and then decreased across all birth cohorts (Figure 1E). It is indicated that the risk of IHD mortality was improved in cohorts born more recently.

### 3.3. Net Drift and Local Drift in Age Groups

Net drift indicated the estimated value of the overall AAPC over the whole study period, whereas local drift indicated the AAPC of each age group (Figure 2). A reduction in IHD mortality due to PM_2.5_ exposure was observed in the overall net drift over the whole study period (−1.64%; 95%CI: −1.80~−1.48), which was more pronounced in women than men (−2.38%, 95%CI: −2.65~−2.10 versus −1.64%, 95%CI: −1.43~−1.11). For the cause of HPM exposure, the overall net drift values for men and women were −8.83% (95%CI: −9.00~−8.66) and −8.99% (95%CI: −9.34~−8.64), respectively, reflecting substantial reductions in IHD mortality. However, IHD mortality attributable to APM exposure for both men and women increased by 1.69% (95%CI: 1.51~1.87) and 1.42% (95%CI: 1.15~1.70), respectively.

Local drift values of IHD mortality attributable to HPM exposure were below 0 in all age groups, indicating an obvious decreasing trend in IHD mortality. The greatest decline was found in men aged 65~69 years (−10.28%; 95%CI: −10.49~−10.06) and women aged 25~29 years (−9.67%; 95%CI: −12.70~−6.55). Conversely, local drift values of IHD mortality attributable to APM exposure were mostly above 0 for both men and women, being initially reduced and then enhanced in men of all ages, and slowly decreased in women.

### 3.4. APC Effects on IHD Mortality Attributable to PM_2.5_ Exposure

Age effects on IHD mortality attributable to PM_2.5_ exposure showed an exponential distribution with a rapidly increased rate in the elderly, which was similar to that due to APM exposure. IHD mortality due to HPM exposure in men and women of all age groups maintained a low level. Caused by APM exposure, IHD mortality in men younger than 50 years was higher than that in women younger than 55 years (Figure 3A–C).

Period effects due to PM_2.5_ exposure showed a slowly decreased trend, suggesting improvements for the whole population across the study period. The improvement was pronounced in women than men from 2010 to 2019, with the corresponding RR of less than 1 for both sexes. The RR of period effects due to APM exposure was gradually enhanced, suggesting progressive increase in risk of IHD mortality for the whole population. It can also be seen that the negative impact of period effects on the whole population was slowly being controlled after 2012, and the positive impact on women appeared earlier than that that of men. Period effects due to HPM exposure sharply decreased during the whole study period, indicating a striking decrease in the risk of IHD mortality due to HPM exposure in the whole population. An overall declining risk for IHD death attributable to PM_2.5_ exposure was presented in successively younger birth cohorts, which was more noticeable in women. It was implied that in recent years, the cohort effect favorably affected the mortality of IHD attributable to PM_2.5_ in the whole population, especially in women. Similar to period effects, cohort effects for IHD mortality due to HPM exposure were markedly declined, with progressive improvement in mortality in those born from 1900 onwards. However, RR of IHD mortality due to APM exposure increased across birth cohorts, and there was some indication of more unfavorable trends in men who were born after 1960 (Figure 3D–I). The age, period, and cohort effects were statistically significant, and the detailed results are listed in Appendix A.

## 4. Discussion

The present study comprehensively assessed temporal trends in IHD mortality attributable to PM_2.5_ exposure, including APM and HPM. A growing number of data have shown that the IHD burden attributable to APM exposure has shifted from a global health concern to an extremely highlighted problem affecting Asia and Africa [1,3], which was also confirmed in our study. From 1990 to 2019, the number of IHD deaths due to APM exposure increased by 4.2 times, and the proportion to all IHD deaths attributable to PM_2.5_ exposure increased from 36.86% to an astonishing 91.72%. Meanwhile, the ASMR due to APM exposure significantly increased, while that attributable to HPM exposure sharply decreased from 12.06/100,000 in 1990 to 0.96/100,000 in 2019. Since 1998, the ASMR due to APM exposure has been gradually higher than that due to HPM exposure, which may be attributed to the rapid urbanization and enhanced income in Jiangsu Province. Jiangsu is a populous region with more industry, where residents generally have better access to cleaner fuels and fewer emissions of PM_2.5_ from household fuels [18]. However, the development of industrialization has exacerbated outdoor air pollution [19,20].

Consistent with previous findings, our study revealed that the IHD mortality due to APM and HPM exposures were higher in men than women from 1990 to 2019. The gender difference could be attributed to the fact that the mortality of IHD is higher in men than women [2,21]. Although men are less likely to be exposed to HPM because of a shorter duration in the kitchen, typical risk factors such as alcohol consumption or smoking are more prevalent in men [22], which may result in a synergist effect with HPM exposure on IHD.

Age-stratified analysis showed that IHD mortality attributable to PM_2.5_ exposure rapidly increased in the elderly, especially in those over 60 years. The aging population has been recognized as a major risk factor for CVD [23]. In addition, a long-term exposure to PM_2.5_ results in the accumulation of harmful particles in the human body [5,24], which may exacerbate the endothelial dysfunction [25], systematic inflammation [26], and hypertension [26] that further increase the risk of IHD mortality. Similarly, a decrease in particulate matter concentrations is associated with IHD mortality decline [27,28]. Thus, the increase in IHD deaths attributable to PM_2.5_ exposure may be resulted from the increased exposure level and aging. Consistently, a relevant study in Singapore reported a substantial influence of PM_2.5_ exposure on cardiovascular mortality in the elderly, rather than the non-elderly population [29]. Therefore, the elderly are a high-risk group that should be concerned for the potential negative influence of PM_2.5_ exposure on IHD. 

Fortunately, IHD mortality due to HPM exposure decreased in all age groups, especially the decrease in local drifts. According to the longitudinal age curves, a flat and relative low mortality of IHD due to HPM exposure was witnessed in the same birth cohort after adjusting for period deviations. Moreover, IHD mortality strikingly declined with time periods and in consecutive birth cohorts, which could be explained by the increase use of clean fuels and sharply decreased emissions of indoor PM_2.5_ exposure from household fuels [30]. It was reported that during 2005–2015, the biomass consumption and urban coal consumption dropped by 58%, resulting in 56% decrease (56% in urban and 45% in rural areas) of the mean HPM exposure in China [18]. The rapid urbanization and improved income were major drivers behind the decrease in solid fuels. Additionally, the popular use of houseplants and air filtration with the economic improvement has been proven to be effective for indoor gas pollutant removal and indoor air quality [31,32]. As a result, the risk of death from IHD attributable to household PM_2.5_ exposure has been significantly reduced. However, the values of net drifts due to APM exposure were positive, and those of local drifts were above 0 in almost all age groups, indicating that IHD mortality due to APM exposure in Jiangsu Province has been worsening, especially in the elderly and men aged 25–40 years. It is considered that occupational exposure by outdoor labor was the fundamental cause, and corresponding management for controlling air pollution was necessary [33].

Our study found that IHD mortality attributable to APM exposure was initially enhanced and then reduced from 1990 to 2019 in all age groups, which was basically consistent with the findings of period effects. Since 2005, the government has implemented stringent control policies to reduce the total amount of air pollutant emissions. In 2013 and 2018, the *National Air Pollution Prevention and Control Action Plan*, mainly in the Beijing–Tianjin–Hebei regions, the Yangtze River Delta, and the Pearl River Deltas [34], and the *Blue Sky Defense War Policy*, a three-year action plan [35], were issued in China, respectively. Therefore, the RR of period effects on IHD mortality due to APM exposure gradually increased from 1990 and then decreased in recent years. Notably, IHD mortality due to APM exposure after adjusting period deviations presented an exponential distribution, indicating the higher risk of IHD mortality due to APM exposure in the elderly. Air pollution was still a challenging issue to be highlighted. More efforts should be made to control air pollution and to strengthen the management of relevant diseases. In addition, high-risk populations such as the elderly and immunocompromised people are the priority to be protected [29,36].

The RR of cohort effects on IHD mortality due to APM exposure was on the rise, especially in people born after 1960s. As we have mentioned, an increasing level of APM exposure alongside the acceleration of industrialization has been a huge threat. Of all risk factors for IHD mortality, APM exposure has been elevated from the 10th risk in 1990 to the 4th in 2019 in Jiangsu Province. In contrast, the risk of HPM exposure has declined from the 10th to the 21st (Appendix A).

Jiangsu Province is located in the Yangtze River Delta region. Rapid social and economic development has greatly prolonged the life expectancy of residents, but it also changed lifestyles. The prevalence of risk factors for CVD has been markedly enhanced, such as hypertension, hypercholesterolemia, obesity, and diabetes mellitus. These risk factors for CVD may interact with ambient particulate matters and increase the IHD mortality, especially in men because of more consumption of alcohol or tobacco [22,37]. Younger birth cohorts are more susceptible to these risk factors. In recent years, the Chinese government has begun to address the challenges of non-communicable diseases (NCDs). A series of strategies and policies have been implemented, including the Medium-to-Long Term Plan for the Prevention and Treatment of Chronic Diseases in Jiangsu (2018–2025) [38] and Healthy Jiangsu 2030 [39]. As expected, a decline trend of period effects in recent years has been observed, suggesting the efficacy of social management. Moreover, these favorable actions to fight against IHD attributable to PM_2.5_ exposure are more effective in recently born cohorts. 

Some limitations should be noted. First of all, our data were obtained from the GBD 2019, which were limited in terms of their insufficient information on non-fatal consequences of IHD [40], underestimated mortality due to a lack of surveillance data on direct provincial cardiovascular events [41], and the joint effects of the two environmental risk factors not being taken into account [5]. Secondly, data on IHD deaths in people younger than 25 years were scant. Thirdly, although period and cohort effects were estimated in this study, our age-period-cohort analysis was based on the estimated cross-sectional data of GBD. Therefore, the relevant hypotheses of this study must be further confirmed in large cohort studies. Fourthly, the APC model only analyzed the effects of age, period, and cohort, and other risk factors should be explored in the future. 

Collectively, we for the first time reported the temporal trend of IHD mortality attributable to PM_2.5_ exposure (APM and HPM) in Jiangsu Province, Yangtze River Delta region, Southeast China, by analyzing data in GBD 2019. Through introducing the APC model, we estimated the independent effects of age, period, and cohort on IHD mortality, within the age-period-cohort model. Our results, to some extent, reflected a relative level in developed provinces in China. Moreover, local drift values reflected the temporal trend of IHD mortality attributable to PM_2.5_ exposure in each age group. The analysis on period and cohort effects contributed to distinguishing the sources of IHD mortality trends in different time periods and birth cohorts. Therefore, our study was able to identify the priority groups for prevention and treatment on the basis of unfavorable trends in IHD mortality attributable to PM_2.5_ exposure in certain age groups, as well as to assist in the tracking of sustainable development goals (SDG) targets.

## 5. Conclusions

IHD mortality due to HPM exposure was strikingly reduced in all age groups in Jiangsu Province, especially in recently born cohorts. However, IHD mortality due to APM exposure has been on the rise within the past three decades, especially in the elderly, after adjusting for period deviations. It was gradually worsening in people born after 1960, as well as in men in recent years, although a favorable period effect was observed from 2015 to 2019. Health strategies, policies, and interventions should be a priority to reduce ambient air pollution, thus reducing the effects of APM exposure on IHD mortality.

## Figures and Tables

**Figure 1 ijerph-20-00973-f001:**
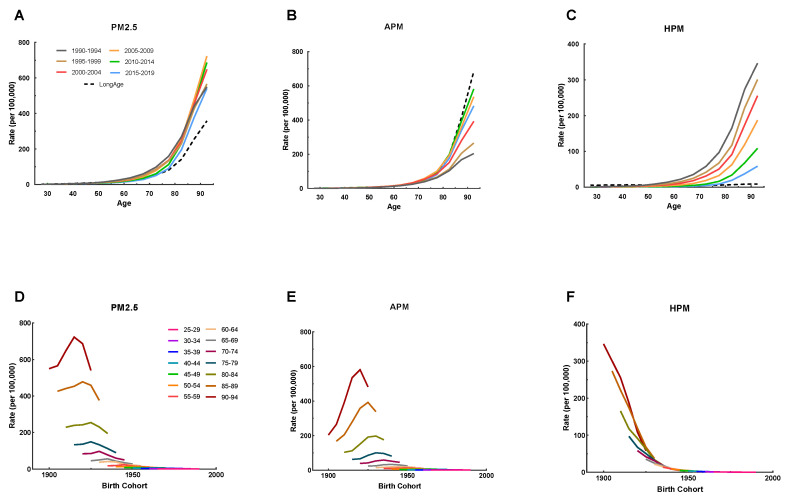
The age-specific mortality rates of IHD attributable to PM_2.5_ (APM and HPM) by period between 1990 and 2019 and cohort-specific mortality rate of IHD attributable to PM2.5 by age group between 1990 and 2019. (**A**–**C**) Survey years were arranged into consecutive 5-year periods from 1990 to 1994 (median, 1992), 1995 to 1999 (median, 1997), 2000 to 2004 (median, 2002), 2005 to 2009 (median, 2007), 2010 to 2014 (median, 2012), and 2015 to 2019 (median, 2017). Longitudinal age curves were estimated by the age-period-cohort model and indicated the expected age-specific rate of IHD mortality due to PM_2.5_ (APM and HPM). (**D**–**F**) The data of IHD mortality attributable to PM_2.5_ (APM and HPM) were arranged into 19 consecutive birth cohorts, including those born from 1898–1902 (median, 1900) to 1988–1992 (median, 1990), and successive 5-year age intervals from 25–29 years (median, 27 years) to 90–94 years (median, 92 years) of age.

**Figure 2 ijerph-20-00973-f002:**
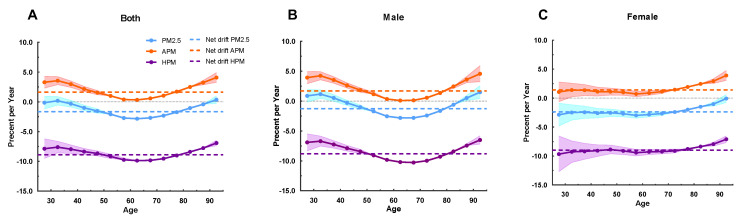
Local drift with net drift values of the mortality for IHD attributable to PM_2.5_ (APM and HPM) in Jiangsu from 1990 to 2019. (**A**) Corresponding to both sexes; (**B**) corresponding to male; (**C**) corresponding to female. The dots and shaded areas denote percentage and their corresponding 95%CI.

**Figure 3 ijerph-20-00973-f003:**
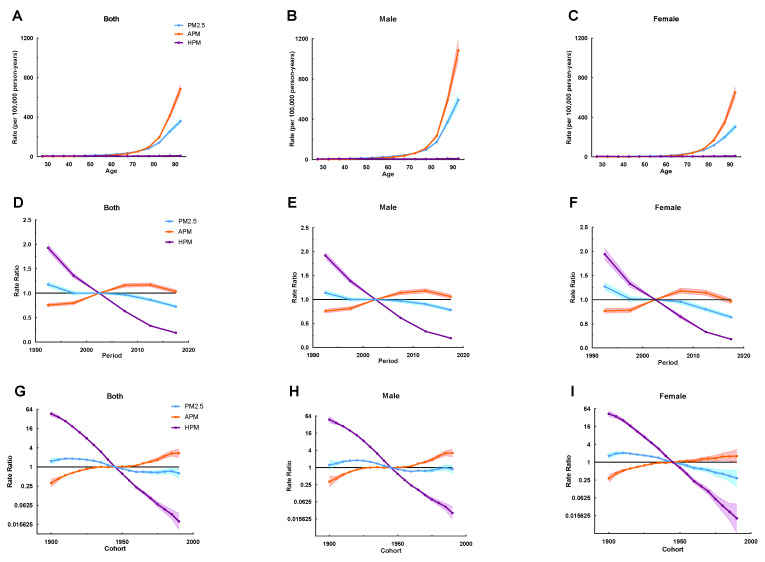
The age-period-cohort (APC) results of IHD attributable to PM_2.5_ (APM and HPM) in Jiangsu from 1990 to 2019. (**A**–**C**) Fitted longitudinal age curves of IHD mortality (per 100,000) attributable to PM_2.5_ (APM and HPM), with (**A**) corresponding to both sexes, (**B**) corresponding to males, (**C**) and corresponding to females. (**D**–**F**) Rate ratio of each period compared with the reference (2000–2004) adjusted for age and nonlinear cohort effects and the corresponding 95% CI, with (**D**) corresponding to both sexes, (**E**) corresponding to males, (**F**) and corresponding to females. (**G**–**I**) Rate ratio of each cohort compared with the reference (cohort 1943–1947) adjusted for age and nonlinear period effects and the corresponding 95% CI, with (**G**) corresponding to both sexes, (**H**) corresponding to males, (**I**) corresponding to females. The dots and shaded areas denote mortality rates or rate ratios and their corresponding 95%CI.

**Table 1 ijerph-20-00973-t001:** Age-standardized mortality rate per 100,000 of IHD attributable to PM_2.5_ exposure (APM and HPM) in 1990 and 2019, and its temporal trends from 1990 to 2019.

Sex	PM_2.5_ Exposure			APM Exposure			HPM Exposure		
	ASMR (1990) (95%UI)	ASMR (2019) (95%UI)	AAPC% (95%CI)	ASMR (1990) (95%UI)	ASMR (2019) (95%UI)	AAPC% (95%CI)	ASMR (1990) (95%UI)	ASMR (2019) (95%UI)	AAPC% (95%CI)
Both	18.90 (15.55~23.33)	11.53 (9.12~14.90)	−1.71 ^a^ (−2.02~−1.40)	6.84 (3.36~11.44)	10.57 (8.24~13.83)	1.45 ^a^ (1.18~1.72)	12.06 (7.99~16.22)	0.96 (0.33~2.16)	−8.27 ^a^ (−8.84~−7.69)
Male	23.32 (18.80~29.66)	15.81 (11.98~20.81)	−1.44 ^a^ (−1.93~−0.95)	9.61 (4.61~15.82)	14.73 (11.10~19.32)	1.38 (0.90~1.86)	13.71 (8.10~19.73)	1.08 (0.36~2.59)	−8.29 ^a^ (−8.64~−7.94)
Female	15.55 (12.01~20.10)	8.64 (6.24~12.86)	−2.06 ^a^ (−2.26~−1.85)	4.89 (2.37~8.56)	7.77 (5.56~11.66)	1.57 ^a^ (1.33~1.81)	10.66 (7.13~14.73)	0.87 (0.31~1.89)	−8.20 ^a^ (−8.74~−7.65)

PM_2.5_: particulate matter < 2.5 μm in aerodynamic diameter; APM: ambient PM_2.5_; HPM: household PM_2.5_; UI: uncertainty interval; ASMR: age-standardized mortality rate; CI: confidential interval; AAPC: average annual percentage change. ^a^ Statistically significant (*p* < 0.05).

## Data Availability

Not applicable.

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
