# Peer review of "Time Trends in Ischemic Heart Disease Mortality Attributable to PM2.5 Exposure in Southeastern China from 1990 to 2019: An Age-Period-Cohort Analysis"

_ijerph, 2023, doi:10.3390/ijerph20020973_

Round 1

Reviewer 1 Report

Wang et al. conducted an age-period-cohort (APC) analysis to explore time trends in ischemic heart disease (IHD) mortality attributable to the exposure to ambient and household particle matter (APM and HPM, respectively) in Jiangsu, China, using estimates from the the Global Burden of Disease (GBD) from 1990 to 2019.

Methods

- Data sources are clearly described; however, there is no description of the statistical approach used to model ACP effects and test them for goodness-of-fit.

Results

- First column in table 1 is not completely shown; however, it can be inferred that rows correspond to male and female ("both"), male, and female from top to bottom.

-  Figures 1-3 are difficult to read due to their small size and low resolution.

- Sentence in lines 163-165 is not clearly stated.

- Please clarify whether lines 172-180 correspond to a note to Figure 1. Does "LongAge" in figures 1A-1C correspond to longitudinal age curves?

- Please clarify whether line 202 corresponds to a note to Figure 2.

- Please clarify whether lines 225-232 correspond to a note to Figure 3.

Discussion

- Overall this section appropriately reflect and contextualize the main findings of the study in connection with their implications for health policy. Limtations are clearly and throughly stated. Finally, conclusion are consistent with the results of the analysis.

Reviewer 2 Report

This study focuses on the temporal effects of PM2.5 on the mortality of ischemic heart disease (IHD) along with the age, period, and birth cohorts in a province of China. Before the final publication, the authors are recommended to address the following comments:

Section 3.1

The authors defined the annual percentage change (APC) with 95% confidence interval for the IHD mortality attributable to PM2.5. The data were collected over 30 years between 1990 and 2019, which means that the authors should consider time series modeling. In Table 1, the comparison between the 1990 and 2019 results may not capture the dynamics of APC. Although the title contains ‘temporal trend’, these two time points may not represent a temporal trend. The authors should include the overall time series plot of APC related to this table, and fit the time trend.  In Table 1, For the male AAPC% of APM exposure, not sure about the statistical significance because the 95% CI contains 1.

Section 3.2

In Figure 1, the split of time periods is arbitrary. The authors are advised to draw a time series plot over the entire period for each group of a cohort. For example, a plot between the age cohort and the IHD mortality attributable to PM2.5 can draw six time series plots (25-34, 35-44, 45-54, 55-64, 65-74, 75-) together to check overall patterns.  Although the authors provide Figure 1, there is no explanation for what this figure demonstrates.  

Section 3.4

The authors just describe the components of the figures. The authors need to elaborate the explanation of results.

Discussion

The authors need to add more explanation for why HPM related IHD mortality decreased. For example, is there any big change in fuel used inside house from wood and coal to natural gas?

The authors should point out the limitations of this study. In particular, this study did not address the temporal dynamics while the comparison of the starting and end points is very weak analysis. Graphical illustrations do not provide any statistical significance either.
